# Randomized Controlled Trials on Renin Angiotensin Aldosterone System Inhibitors in Chronic Kidney Disease Stages 3–5: Are They Robust? A Fragility Index Analysis

**DOI:** 10.3390/jcm11206184

**Published:** 2022-10-20

**Authors:** Ivana Capuano, Pasquale Buonanno, Eleonora Riccio, Antonio Bianco, Antonio Pisani

**Affiliations:** 1Department of Public Health, University of Naples “Federico II”, 80131 Naples, Italy; 2Department of Neurosciences, Reproductive and Odontostomatological Sciences, University of Naples “Federico II”, 80131 Naples, Italy; 3Institute for Biomedical Research and Innovation, National Research Council of Italy, 80125 Palermo, Italy; 4Interdepartmental Research Center for Arterial Hypertension and Associated Pathologies (CIRIAPA)-Hypertension Research Center, University of Naples “Federico II”, 80131 Naples, Italy

**Keywords:** chronic kidney disease 1, renin-angiotensin-aldosterone system inhibitors, fragility index, randomized controlled trials, robustness

## Abstract

Inhibition of the renin-angiotensin-aldosterone system (RAAS) is broadly recommended in many nephrological guidelines to prevent chronic kidney disease (CKD) progression. This work aimed to analyze the robustness of randomized controlled trials (RCTs) investigating the renal and cardiovascular outcomes in CKD stages 3–5 patients treated with RAAS inhibitors (RAASi). We searched for RCTs in MEDLINE (PubMed), EMBASE databases, and the Cochrane register. Fragility indexes (FIs) for every primary and secondary outcome were calculated according to Walsh et al., who first described this novel metric, suggesting 8 as the cut-off to consider a study robust. Spearman coefficient was calculated to correlate FI to *p* value and sample size of statistically significant primary and secondary outcomes. Twenty-two studies met the inclusion criteria, including 80,455 patients. Sample size considerably varied among the studies (median: 1693.5, range: 73–17,276). The median follow-up was 38 months (range 24–58). The overall median of both primary and secondary outcomes was 0 (range 0–117 and range 0–55, respectively). The median of FI for primary and secondary outcomes with a *p* value lower than 0.05 was 6 (range: 1–117) and 7.5 (range: 1–55), respectively. The medians of the FI for primary outcomes with a *p* value lower than 0.05 in CKD and no CKD patients were 5.5 (range 1–117) and 22 (range 1–80), respectively. Only a few RCTs have been shown to be robust. Our analysis underlined the need for further research with appropriate sample sizes and study design to explore the real potentialities of RAASi in the progression of CKD.

## 1. Introduction

Inhibition of the renin-angiotensin-aldosterone system (RAAS) is a key strategy in the treatment of hypertension, cardiovascular and renal diseases. In particular, current guidelines recommend the use of RAAS inhibitors (RAASi) as first-line antihypertensive therapy for patients with proteinuric chronic kidney disease (CKD) and diabetic nephropathy on the basis of randomized controlled trials (RCTs) [1]. These studies demonstrated that RAASi are superior to other antihypertensive drugs in delaying the nephropathy progression to end stage renal disease (ESRD), providing a significant renal and cardiovascular protection for CKD patients in the earlier stages [2,3]. Unfortunately, only a few RCTs are available on patients with advanced CKD 3–5 to define whether angiotensin converting enzyme inhibitors (ACEIs) or angiotensin receptor blockers (ARBs) can effectively delay ESRD and reduce mortality in this population [4]. Fragility index (FI) is a novel parameter proposed by Walsh et al. to challenge the robustness of clinical studies. It can be used to test RCTs with dichotomous outcomes and 1:1 randomization of patients in control and experimental groups. FI is defined as the minimum number of patients of the experimental group that should pass from event to nonevent class in order to make a statistically significant difference between groups nonsignificant (i.e., to produce a *p* value lower than 0.05). A small FI indicates a more fragile study because its statistical significance is based on a small number of patients who experienced the treatment effect [5]. FI has been used so far to explore robustness in many clinical fields such as cardiology, urology, critical care, orthopedics, neurosurgery, and nephrology [6,7,8,9,10,11,12]. We performed the present study to explore this novel metric and assess the fragility of RCTs published on renal and cardiovascular outcomes of RAASi.

## 2. Materials and Methods

### 2.1. Participants

We included studies investigating patients aged ≥ 18 years with CKD 3–5 (defined as glomerular filtration rate (GFR) < 60 mL/min/1.73 m^2^ or elevated serum creatinine level > 1.2 mg/dL) not under dialysis treatment.

### 2.2. Interventions and Comparators

Patients in the intervention group were treated with ACEIs and/or ARBs and they were compared to patients treated with placebo and/or control therapy, defined as therapy with drugs other than RAASi.

### 2.3. Systematic Review Protocol

We performed a systematic literature search to identify studies addressing our topic according to the recommendations of the Cochrane Collaboration, as well as the guidelines of the Preferred Reporting Items for Systematic Reviews and Meta-Analyses Statement for Reporting Systematic Reviews (PRISMA).

### 2.4. Data Sources and Search Strategy

We searched for RCTs in MEDLINE (PubMed), EMBASE databases, and the Cochrane register. The terms “chronic kidney disease”, “randomized controlled trial”, and all spellings of ACEi and ARBs were used individually or in different combinations to screen the literature. We also reviewed the references lists of identified trials to detect other potential studies to be included in our analysis. Missing data or more detailed information were requested from authors when not available in their publications. We included only articles in English without sample size restrictions. Articles were evaluated by two independent reviewers (P.B. and I.C.) and disagreements were resolved by discussion or by a third investigator (E.R.). This study was registered on SRDR+ (https://srdrplus.ahrq.gov/ (accessed on 10 October 2020)).

### 2.5. Study Selection and Data Extraction

After screening titles and abstracts, 45 studies reporting renal and cardiovascular outcomes of RAASi in patients with non-dialysis chronic kidney disease stages 3–5 were considered (Figure 1). We considered both studies focusing on patients suffering from CKD and studies conducted on the general population; in the latter case, we extracted the data about CKD patients. The full text of each article was reviewed by two authors and 22 articles were included in the present paper [13,14,15,16,17,18,19,20,21,22,23,24,25,26,27,28,29,30,31,32,33,34]. COPE and IDNT studies included three arms in their design, so we considered each of them as two different studies comparing two groups. In particular, the COPE study included a group treated with ARB, one group with beta-blockers, and one group with thiazide diuretics, and we considered two comparisons: ARB vs. beta-blockers and ARB vs. thiazide diuretics [31]. The IDNT study included a group treated with ARB, an active control (amlodipine), and a placebo group, and we considered the comparison ARB vs. active control and ARB vs. placebo [32]. Consequently, we had 24 comparisons from 22 articles. Twenty one studies were excluded for the following reasons: one study compared dual block of RAAS versus monotherapy with ACEi or ARB [35]; one study compared ARBs versus ACEi [36]; two were not in English language [37,38]; one did not specify the number of events [39]; one was without 1:1 randomization [40]; fifteen had no dichotomous outcome [41,42,43,44,45,46,47,48,49,50,51,52,53,54,55]. The data about clinical outcomes of RAASi treatment derived mostly from non-nephrological journals. The total population consisted of 80,455 individuals. We extracted information using standard data extraction forms, which included baseline patient characteristics, intervention, dose of drug, follow-up duration, outcome events, and adverse events. These data were extracted from both studies which were conducted only for CKD 3–5 patients and studies which included subgroup populations with CKD 3–5 at baseline. We used standard criteria (Cochrane risk of bias tool) to assess the inherent risk of bias in the trials, as shown in Appendix A. Two investigators (I.C. and P.B.) independently undertook data extraction and quality assessment using a standardized approach. Any disagreements between the two investigators were resolved by consultation with a third reviewer (E.R.).

### 2.6. Risk of Bias

Sources of bias were divided into five domains (i.e., randomization, deviation from intended interventions, accounting for missing data, measurements, and selectively reporting results) and evaluated by the Cochrane Risk of Bias Tool 2.0. According to this tool, studies have been classified into three categories: low risk, some concerns, and high risk of bias. Studies with all the five domains at low risk were overall considered at low risk; one domain with some concerns led the study to be classified as affected by some concerns of bias. Trials with two domains with some concerns, or one domain with high risk of bias, were considered at high risk.

### 2.7. Data Analysis: Fragility Index Calculation

Fragility index was determined using the method of Walsh et al. for both primary and secondary outcomes: we first determined *p* value through Fisher’s exact test and then we progressively moved patients in the experimental arm from the event to the nonevent group until *p* value became higher than 0.05. The minimum number of patients to make *p* value exceed 0.05 represents FI [5]. Correlations between FI and *p* value and sample size were investigated using the Spearman rank test. In order to normalize the FI on the basis of the sample size, we calculated the fragility quotient (FQ), which is the result of the ratio between the FI and the total sample size.

## 3. Results

### 3.1. Characteristics of Trials

Table 1 summarizes the RCTs’ characteristics. Sample size considerably varied among the studies (median: 1693.5, range: 73–17,276). The median follow-up was 38 months (range 24–58). Fifty percent of the studies had a statistically significant *p* value and 14% of them presented a *p* value lower than 0.001. Fifty percent of the trials focused their attention only on CKD patients whereas the others investigated the effects of RAASi in the general population. The majority of studies were blinded, but only 12.5% of trials showed an adequate allocation concealment. Data were elaborated by intention-to-treat analysis in 54.5% of the included RCTs.

### 3.2. Fragility Index

Figure 2 shows the frequency of fragility index values for primary and secondary outcomes (panels A and B, respectively); for both primary (Figure 2A) and secondary (Figure 2B) outcomes, most of the studies showed a fragility index of 0. The overall median of primary outcomes fragility index was 0 (range 0–117) and for secondary outcomes it was also 0 (range 0–55). Medians of FI for primary and secondary outcomes with a *p* value lower than 0.05 were 6 (range: 1–117) and 7.5 (range: 1–55), respectively. Medians of FI for primary outcomes with a *p* value lower than 0.05 in CKD and no CKD patients were 5.5 (range 1–117) and 22 (range 1–80), respectively; this difference between medians of FI in CKD and no CKD populations was confirmed also for secondary outcomes with *p* value lower than 0.05: 5.5 (range 1–33) and 13.5 (range 1–55), respectively (Table 2) (see Appendix A for details about the fragility index for studies with a *p* value lower than 0.05; Appendix A report the fragility index for primary and secondary outcomes, respectively). The median (and range) of FQ for all the studies in CKD and no CKD patients with a statistically significant primary outcome (i.e., *p* < 0.05) were 7.9‰ (0.7–36‰) and 2.7‰ (0.3–11.1‰), respectively. A similar result was found for secondary outcomes: the median and range of FQ for studies with statistically significant results in CKD and no CKD patients were 8.3‰ (2.0–31.8‰) and 4.6 (1.2–19.1‰), respectively. Figure 3 reports the scatter plots of FI versus *p* value for both primary and secondary outcomes, which show a strong inverse correlation (rs = −0.7902 and −0.7196, respectively; *p* < 0.0001); a strong positive correlation was also found between FI and studies’ sample size for primary and secondary outcomes (rs = 0.6315 and 0.6013, respectively; *p* < 0.01).

## 4. Discussion

The aim of this study was to explore the robustness of RCTs investigating clinical outcomes related to RAASi treatment in CKD stages 3–5. We found that sample size and follow up period showed great variability among the selected studies. Furthermore, only 50% of them reported significant primary and secondary outcomes. Fragility indexes of studies with significant *p* values were very low for both primary and secondary outcomes in CKD patients, whereas a higher FI was registered in the non-CKD population. In addition, FI was shown to strongly correlate with *p* value and sample size.

Our results agree with the previous observation of Strippoli GFM et al. which underlined the lack of adequate trial methods and study design in nephrological research and a poor number of RCTs compared to other internal medicine specialties; in particular, we found strikingly poor attention to allocation concealment and little use of intention-to-treat analysis [3]. Allocation concealment is often confused with blinding. Concealment is a practice to make the researcher unaware of the sequence of patient allocation so he cannot know the group to which the next patient will be assigned. Concealment avoids conscious or unconscious exclusion of patients who could not respond to researcher expectations if assigned to a particular experimental group. On the other hand, blinding refers to the unawareness of patients, care providers, and sometimes data analyzers, about the intervention administered to the participants; this practice reduces placebo effect in single blinded studies and also the misinterpretations of results by clinicians in double blinded trials [56].

Intention-to-treat analysis is another essential practice to avoid overestimations of treatment; this study design analyzes results, holding each participant and the corresponding outcome in his original group regardless of the treatment he is actually receiving, and regardless of withdrawal or deviation from the protocol. It allows researchers to preserve randomization, it avoids reduction of study power if a large number of patients withdraw from the protocol, and it prevents the exclusion of patients who are supposed not to respond to researcher expectations during the study [57].

One of the most important findings of our study is that the median FI for RCTs investigating RAASi treatment in CKD stages 3–5 population is very low, thus underlining the weakness of these trials. There is not a cut-off value for FI but 8 is generally deemed as the minimum value to consider a study robust. As with FI, there is no cutoff for FQ to distinguish robust vs. fragile studies, but it is intuitive that if only about 8 patients in a total sample size of 1000 patients (as in the case of primary outcome in studies investigating the role of RAASi in CKD patients) are sufficient to turn a significant outcome into a nonsignificant result, those studies cannot be considered very robust and, consequently, their findings should be considered with caution. On the contrary, RCTs including no CKD patients far exceed the FI cut off value of 8. This finding could be explained by the complexity of patients suffering from CKD, particularly in the advanced stages which are characterized by the presence of many comorbidities [58]. Our results did not differ from similar studies investigating FI in the RCTs of other specialties; Evaniew N. et al. and Ridgeon EE et al. reported a median FI of 2 in the field of spine surgery and critical care, respectively, while Narayan VM et al. calculated a median FI of 3 in urological RCTs [6,8,10]. Shochet LR analyzed nephrological RCTs published in five nephrology journals and five general journals for a period ranging from 2005 to 2014 and found that the 127 studies meeting the inclusion criteria had a median FI of 3 [12]. Our study mirrors the abovementioned results with a median FI of 5.5 both for primary and secondary outcomes. While the majority of studies indicate a lack of robustness for RCTs in many medical fields, fragility index values in cardiology and internal medicine are often higher than 8 [59]. It is noteworthy that we found robust FIs in studies with no CKD patients and that these findings come from studies conducted by cardiologists and specialists in internal medicine.

Our work underlined the need for studies enrolling an adequate number of patients; in fact, sample size was strongly correlated with FI, so larger studies were more robust. This finding is not obvious, as other works on fragility index did not find any relationship between FI and sample size [11]; however, the larger the sample size is, the smaller are the differences which can be significant in the statistical analysis, thus producing significant findings even with little difference in the number of events between the intervention and control groups. Our study showed that by increasing the sample size, the differences in events in the treatment group are augmented, thus producing a higher FI.

## 5. Conclusions

Our study demonstrated that the results of RCTs investigating the favorable renal and cardiovascular outcomes of RAASi in patients affected by CKD 3–5 need to be read with extreme caution. Further research with appropriate sample sizes and study design is needed to explore the real potentialities of RAASi in the progression of CKD.

## Figures and Tables

**Figure 1 jcm-11-06184-f001:**
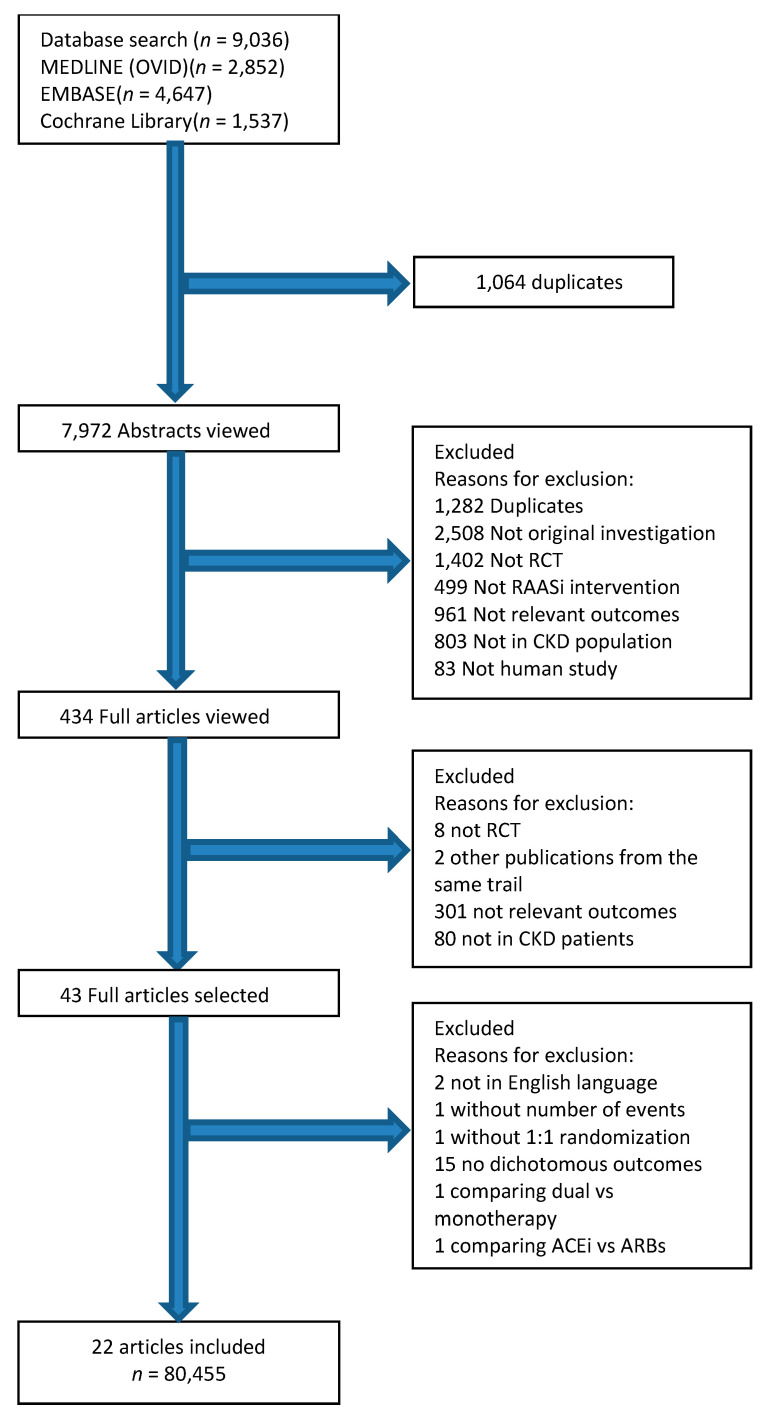
Flow diagram.

**Figure 2 jcm-11-06184-f002:**
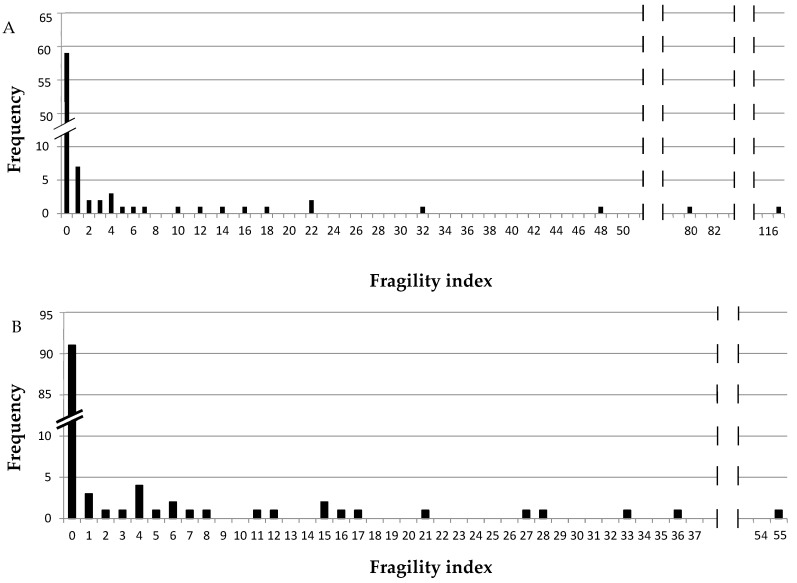
Frequency of fragility index values. Panels (**A**,**B**) show the absolute frequency of fragility index values for primary and secondary outcomes, respectively.

**Figure 3 jcm-11-06184-f003:**
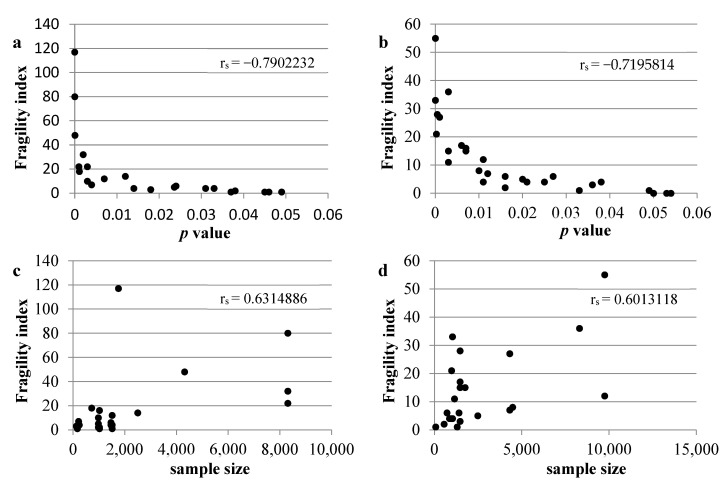
Scatter plots of fragility index. Fragility index versus *p* value for primary (**a**) and secondary (**b**) outcomes; fragility index versus sample size for primary (**c**) and secondary (**d**) outcomes.

**Table 1 jcm-11-06184-t001:** Characteristics of included trials.

Trial Characteristics	Studies (*n*)
**Sample size**, median (min–max)	1693.5 (73–17,276)
**Follow-up** (months), median (min–max)	38 (24–58)
**ACEi vs. placebo**	8
**ARB vs. placebo**	4
**ACEi vs. active control**	5
**ARB vs. active control**	6
**RAASi vs. active control or ACE-I vs. no treatment**	1
**Primary endpoint**	
*p*-value	
>0.05	10
0.01–0.05	8
0.001–0.01	2
<0.001	4
**Secondary endpoints**	
** *All-cause mortality* **	
Reported *p*-value < 0.05	3
Total	8
** *Cardiovascular mortality* **	
Reported *p*-value < 0.05	2
Total	5
** *All-cause hospitalization* **	
Reported *p*-value < 0.05	2
Total	2
** *CV hospitalization* **	
Reported *p*-value < 0.05	2
Total	2
** *Major CV events* **	
Reported *p*-value < 0.05	4
Total	6
** *Major cerebrovascular events* **	
Reported *p*-value < 0.05	2
Total	5
** *New or worsening nephropathy* **	
Reported *p*-value < 0.05	1
Total	2
**Allocation concealment**	
None or unknown	21
Yes	1
**Intention-to-treat analysis**	
Yes	12
No or unclear	10
**Blinding to treatment**	
Yes	14
No or unclear	8
**Journals**	
Nephrological	5
General medical	17
**Type of population**	
CKD	10
CKD and no CKD	12

**Table 2 jcm-11-06184-t002:** Median of fragility index for primary and secondary outcomes with *p* value lower than 0.05 stratified by CKD and no CKD condition.

Outcome	FI Median (Range)
**Primary**	
CKD	5.5 (1–117)
no CKD	22 (1–80)
**Secondary**	
CKD	5.5 (1–33)
No CKD	13.5 (1–55)

## Data Availability

The data underlying this article are available in the article and in its online Appendix A.

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
