# Peer review of "Randomized Controlled Trials on Renin Angiotensin Aldosterone System Inhibitors in Chronic Kidney Disease Stages 3–5: Are They Robust? A Fragility Index Analysis"

_jcm, 2022, doi:10.3390/jcm11206184_

Round 1

Reviewer 1 Report

A clear refreshing , open-minded rigorous analysis of literature on protective effect of drugs acting on renin-angiotensin system in chronic kidney disease showing the weakness of the controlled trials in this population and the need of a reappraisal of our intellectual confort zone.

Reviewer 2 Report

The authors indicated a fragility index analysis to clarify the real potentialities of renin-angiotensin aldosterone system (RAAS) in chronic kidney disease (CKD) stages 3-5.  Therefore, this manuscript is important.  However, there are some problems in this manuscript.

(1)  This study is performed to compare between the patients who were treated with RAAS inhibitors and those who were treated with placebo and / or antihypertensive drugs other than RAAS.  However, the authors included the study that was compared between dual therapy and monotherapy of RAAS inhibitors and the study that was compared between angiotensin converting enzyme inhibitor and angiotensin II receptor blocker.  The authors should exclude the studies and analyze the remaining data again.

(2)  The authors described “5 Correlations between FI and p value and sample size were investigating…” in page 4, line 140.  I think that 5 is a mistake.  Therefore, the authors should exclude 5 in the sentence.

(3)  It is unclear how the primary outcomes are defined.  The authors should indicate

it as secondary outcomes in Table 1.

(4)  The median of sample size is written as 1329.5.  On the other hand, the data is

described as 1513 in Table 1.  The authors should correct the data adequately.

(5)  There are no Table S1 and Figure 3 in this manuscript.  The authors should

include Table S1 and Figure 3 in this manuscript.

(6)  The authors described “no CKD population was confirmed also for secondary

outcomes with p value lower than 0.5” in page 5, lines 168 to 169.  However, I think that 0.5 is a mistake and 0.05 is correct.  The authors should correct the mistake adequately.

Round 2

Reviewer 2 Report

    The authors responded my concerns adequately.  However, there are some mistakes in Table 1.  

    The sum of the numbers in Table 1 do not match 22. 

    ACEi vs placebo: 8, ARB vs placebo: 4, ACEi vs active control: 5, ARB vs active control: 6 and RAASi vs. active control or ACE-I vs. no treatment: 1 (total 24).

    In addition, Primary endpoint: p-value >0.05: 12, 0.01-0.05: 8, 0.001-0.01: 2, <0.001: 4 (total 26).

    The authors should revise the data adequately in Table 1.

Reviewer 3 Report

The authors satisfactorily responded to my concern. 

Author Response

Thanks to the reviewer for appreciating our work and for the time he spent to ameliorate our article.